# Complex Role of Regulatory T Cells (Tregs) in the Tumor Microenvironment: Their Molecular Mechanisms and Bidirectional Effects on Cancer Progression

**DOI:** 10.3390/ijms25137346

**Published:** 2024-07-04

**Authors:** Yu Wang, Jiazhou Li, Shingo Nakahata, Hidekatsu Iha

**Affiliations:** 1Department of Microbiology, Oita University Faculty of Medicine, Yufu 879-5593, Japan; m21d9024@oita-u.ac.jp; 2Division of Biological Information Technology, Joint Research Center for Human Retrovirus Infection, Kagoshima University, Kagoshima 890-8544, Japan; k2830879@kadai.jp; 3Division of HTLV-1/ATL Carcinogenesis and Therapeutics, Joint Research Center for Human Retrovirus Infection, Kagoshima University, Kagoshima 890-8544, Japan; snakahata@kufm.kagoshima-u.ac.jp; 4Division of Pathophysiology, The Research Center for GLOBAL and LOCAL Infectious Diseases (RCGLID), Oita University, Yufu 879-5593, Japan

**Keywords:** regulatory T cells (Tregs), cancer, tumor microenvironment (TME), chemokine

## Abstract

Regulatory T cells (Tregs) possess unique immunosuppressive activity among CD4-positive T cells. Tregs are ubiquitously present in mammals and function to calm excessive immune responses, thereby suppressing allergies or autoimmune diseases. On the other hand, due to their immunosuppressive function, Tregs are thought to promote cancer progression. The tumor microenvironment (TME) is a multicellular system composed of many cell types, including tumor cells, infiltrating immune cells, and cancer-associated fibroblasts (CAFs). Within this environment, Tregs are recruited by chemokines and metabolic factors and impede effective anti-tumor responses. However, in some cases, their presence can also improve patient’s survival rates. Their functional consequences may vary across tumor types, locations, and stages. An in-depth understanding of the precise roles and mechanisms of actions of Treg is crucial for developing effective treatments, emphasizing the need for further investigation and validation. This review aims to provide a comprehensive overview of the complex and multifaceted roles of Tregs within the TME, elucidating cellular communications, signaling pathways, and their impacts on tumor progression and highlighting their potential anti-tumor mechanisms through interactions with functional molecules.

## 1. Introduction

Malignant tumors are distinguished by their capacity for unlimited proliferation, invasion, and metastasis. These characteristics are not solely attributed to genomic variations but are also closely influenced by the tumor microenvironment (TME). The TME plays a crucial role in tumorigenesis by facilitating interactions between tumor cells and surrounding cells via the circulatory and lymphatic systems, influencing cancer development and progression [1]. Malignant tumors often exhibit invasive growth into surrounding tissues, unlike benign tumors, which are typically encapsulated and non-destructive [2,3]. Additionally, malignant tumors tend to be larger in size, have irregular borders, and display more invasive behavior compared with benign tumors [3]. The complexity of tumors extends beyond cancer cells to include various stromal cells within the TME [4]. Tumor-infiltrating lymphocytes (TILs) constitute a crucial component of the TME, significantly influencing cancer progression. They are pivotal in immune surveillance mechanisms and comprise various cell types such as natural killer (NK) cells, CD8^+^ cytotoxic T cells, and CD4^+^ helper T type 1 (Th1) cells. Additionally, regulatory cells modulate immune responses to prevent excessive inflammation. These regulatory cells include tumor-associated macrophages, tolerant dendritic cells, myeloid-derived suppressor cells (MDSCs), and regulatory T cells (Treg cells).

Treg cells, known for their regulatory role and immunosuppressive function within the TME, play a pivotal role in maintaining immune homeostasis [5]. They are widely recognized as significant impediments to effective anti-tumor immune responses, thus rendering them pivotal targets for tumor immunotherapeutic interventions [6]. As a subpopulation originally designated for maintaining immune tolerance, Treg cells are distributed around numerous vital organs to mitigate potential organ damage caused by the hyperactive T cells [7]. However, in individuals with tumors, the immune heterogeneity exhibited by Treg cells presents a formidable challenge to the immune system’s ability to recognize and effectively eliminate tumor cells [8]. Consequently, targeting Treg cells to reshape the immunogenic TME has become an increasingly attractive cancer treatment strategy [5]. Simultaneously, several clinical studies on malignancies such as head and neck squamous cell carcinoma (HNSCC), follicular lymphoma, colorectal cancer, and gastric cancer have demonstrated a positive association between Treg infiltration and enhanced survival rates among afflicted patients [9,10,11,12,13,14]. The intricate regulatory function of Treg cells within the tumor environment underscores the inaccuracy of oversimplifying their role as solely conducive to tumor progression. This article provides a comprehensive overview of Treg cells, covering their types, metabolic modes, and homing mechanisms while highlighting their potential anti-tumor effects. These effects encompass not only the inhibition of tumor-promoting inflammatory responses but also the enhancement of anti-tumor immunity.

## 2. Phenotype and Classification of Treg

The activation of CD4^+^ T cells is accompanied by growth, proliferation, and differentiation into functional effector T cells (Teff) or regulatory T cells (Treg) subsets, which play either stimulatory or inhibitory roles in immunity and inflammation, respectively [6,15,16]. The transition from quiescent naive T cells to Teff or Treg involves precise programming of the activated CD4^+^ T cells [5,6,17].

Human Treg cells exhibit high phenotypic and functional heterogeneity, leading to diverse classification. Currently, two predominant classification methods are widely utilized: the functional subtype classification method and the Th-like Treg subtype classification, based on cytokine expression patterns. However, in recent years, with the widespread adoption of single-cell RNA sequencing (scRNA-seq) analysis, there has been a notable trend towards exploring Treg heterogeneity at the molecular level [18,19,20]. This approach has facilitated the identification of key factors involved in Treg differentiation, prompting researchers to update existing classification systems or develop novel subpopulation classification frameworks (Table 1 and Table 2) [21].

### 2.1. The Classification of CD4^+^ Treg Subpopulation Based on CD45RA

Based on the expression of CD45RA, CD4^+^ Treg subpopulations are classified according to a method proposed by Sakaguchi et al. in 2009 [22]. CD45RA is an isoform of the CD45 protein that is expressed on the surface of T cells and serves as a marker for naive or resting T cells. Typically, CD45RA is expressed on the naive T cells that have not yet encountered their specific antigen, which is prominently found on T cells in cord blood and on unresponsive T cells to recall antigens [23,24]. This classification delineates three main subsets: naive Treg or resting Tregs (rTreg), characterized by Foxp3^low^ CD25^low^ CD45RA+ expression; effector Treg (eTreg), characterized by Foxp3^high^ CD25^high^ CD45RA^−^ expression; and non-Treg, characterized by Foxp3^low^ CD25^low^ CD45RA^−^ expression [22,25,26]. rTreg cells possess certain immunosuppressive functions and expresses markers of naive cells such as CCR7 and CD62L, which can differentiate into eTreg upon antigen stimulation. eTreg cells, also known as activated Treg, exhibit strong proliferation and immunosuppressive functions but are more prone to apoptosis. Non-Treg cells exhibit significant functional heterogeneity. The CD127^+^ subset resembles conventional T cells, while the CD127^−^ subset can secrete pro-inflammatory cytokines such as IFN-γ and IL-17.

### 2.2. Classification of CD4^+^Treg Subpopulations Based on Cytokine Secretion and Transcription Factor Expression

In 2017, Halim et al. introduced a classification of CD4^+^ Tregs into distinct subtypes termed Th-like Tregs, categorized according to their cytokine secretion profiles and transcription factor expression [27,28]. Each Th-like Tregs subset exhibits transcription factor expression and cytokine secretion patterns resembling those of the corresponding Th cells [27,29].

Th1-like Treg cells are characterized by the expression of CCR4^+^ CCR6^+^ CXCR3^+^ and secretion of IFN-γ and TNF-α [30,31,32]. Th1-like Treg cells upregulate the transcription factor T-bet and other Th1 cell markers, such as CCR5. These cells display upregulated expression of the transcription factor T-bet and other Th1 cell-associated markers like CCR5. Utilizing an in vitro model stimulated with IL-12, Th1-like Treg cells manifest activated PI3K and Akt kinase pathways, as well as the transcription factor FoxO, which contributes to the modulation of their IFN-γ secretion and, consequently, their diminished suppressive capability [32].

Th2-like Treg cells: In patients with systemic sclerosis, an increased frequency of Th2-like Treg cells is observed in the skin but not in peripheral blood [33]. These cells are characterized by the expression profile of CCR4^+^ CCR6^−^ CXCR3^−^ and exhibit secretion of IL-2 and IL-5, along with heightened secretion of IL-4 and IL-13, as well as upregulation of Gata-3 and IRF-4 [34].

Th17-like Tregs represent a minority subset of human peripheral Treg cells capable of producing IL-17 and upregulating the transcription factor RORγt (RORC) in vitro while maintaining their suppressive functionality. Characterized by the expression of CCR4^+^ CCR6^+^ CXCR3^−^, these cells secrete IL-17A/IL-17F and are stimulated by IL-1β and IL-6 [31,32,35]

### 2.3. Classification of CD4^+^Treg Subgroups Based on Whether They Stably Express Foxp3

Tregs display notable heterogeneity in both phenotype and function. Researchers classify Tregs into three categories based on the expression of Foxp3 and its suppressive effects. Stable Tregs: These cells express Foxp3 along with other stable markers such as Nrp1 and Helios [36,37]. They possess immunosuppressive function and are the most common subset of Tregs, constituting a significant portion of tumor-infiltrating Tregs. Unstable Tregs: This subset exhibits low or absent Foxp3 expression while secreting IFN-γ and IL-17. They contribute to promoting anti-tumor immunity. Fragile Tregs: These cells express Foxp3 while also secreting IFN-γ. They possess partial anti-tumor effects [38].

### 2.4. Classification According to the Origin

Natural regulatory T cells (nTregs), also known as thymus-derived Tregs (tTregs), are generated in the thymus during the negative selection process through interactions with medullary dendritic cells and exposure to self-antigens presented by thymic epithelial cells [39,40]. Once activated, they are transported to the periphery, where they exhibit suppressive activity against self-antigens through cell contact-dependent mechanisms involving granule enzyme B/perforin or Fas/FasL pathways. This subset of Tregs plays a crucial role in maintaining peripheral tolerance [41,42]. nTregs are identified as CD4^+^ T cells expressing high levels of IL-2Ra (CD25) and low levels of IL-7Ra chain (CD127), characterized by the phenotype CD4^+^ CD25^+^ Foxp3^+^ Helios^+^ CTLA4^+^ Nrp1^+^. Their effector molecules include transforming growth factor-β (TGF-β), IL-10, CTLA-4, IL-35, LAG3, and LAP [41,43,44,45].

**Table 1 ijms-25-07346-t001:** Phenotype and characteristic of Tregs under various classification.

Classification	Cell Subtype	Phenotype	Characteristic
CD45RA^+/−^ [22,25,26]	Resting Tregs (rTreg)	CD45RA^+^, Foxp3^low^ CD25^low^, CCR7^+^, CD62L^+^	Weak immunosuppressive activity
	Effector Treg (eTreg)	CD45RA^−^, Foxp3^high^ CD25^high^	Strong proliferation and immunosuppressive activity
	Non-Treg	CD45RA^−^, Foxp3^low^, CD25^low^, CD127^+^	Weak immunosuppressive activity
		CD45RA^−^, Foxp3^low^ CD25^low^, CD127^−^	Low immunosuppression and producing pro-inflammatory cytokines IFN-γ and IL-17
Th-like Tregs [27,28,29,30,31,32,33,34,35]	Th1-like	CCR4^+^, CCR6^+^, CXCR3^+^, CCR5, CD127^low^, T-bet^high^	Producing IFN-γ
	Th2-like	CCR4^+^, CCR6^−^, CXCR3^−^, Gata-3^high^, IRF4^high^	Producing IL-2, IL-4, IL-5 and IL-13
	Th17-like	CCR4^+^, CCR6^+^, CXCR3^−^, RORγt^high^	Producing IL-17
Foxp3 expression [36,37,38]	Stable Tregs	CD4^+^, CD25^+^, Foxp3^+^, Helios^+^, CTLA4^+^, Nrp1^+^, TIGIT^+^	Producing IL-10, TGF-β and IL-35
	Unstable Tregs	CD16^+^, CD56^+^, Foxp3^low/−^	Producing IFN-γ and IL-17
	Fragile Tregs	Foxp3^+^, Nrp1^−^	Producing IFN-γ
Origin [37,38,39,40,41,42,43,44,45,46,47,48,49,50,51,52]	Thymus-derived T cells (nTreg/tTreg)	CD4^+^, CD25^+^, Foxp3^+^, Helios^+^, CTLA4^+^, Nrp1^+^	Suppressive activity against self-antigens
	Peripheral Treg (pTreg)	CD4^+^, CD25^+^, Foxp3^+/−^, Helios^−^, CTLA4^+^, Nrp1^−^	Immune responses to exogenous antigens
	Induced Tregs (iTregs)	CD4^+^, CD25^+/−^, Foxp3^+/−^, Helios^+^, CTLA4^+^, Nrp1^+^	Induced in vitro

In contrast to tTregs, peripheral Treg (pTregs) play a significant role in preventing and addressing immune responses to exogenous antigens. Under certain conditions, peripheral naïve T cells or conventional T cells can be converted into pTreg cells through the induction of TGF-β, IL-2, or T cell receptor (TCR) [46]. It is worth noting that retinoic acid also promotes pTregs generation [47]. Mucosal dendritic cells, particularly CD103^+^ DCs, induce the generation of Foxp3^+^ Tregs through TGF-β and retinoic acid [48]. Compared with nTregs, pTregs exhibit unstable Foxp3 expression and lack a complete epigenetic foundation of Tregs, thus demonstrating unstable functionality [49].

Induced Tregs (iTregs) refer to Tregs induced in vitro, which also exhibit instability [50,51,52]. In terms of differentially expressed genes, IKZF2 and Nrp1 are typically recognized as markers for tTregs, but not for iTreg and pTreg cells [53]. The iTreg subset encompasses both Foxp3^+^ iTregs and Foxp3^−^ iTregs. Foxp3^+^ iTregs are characterized by the phenotype CD4^+^ CD25^+^ Foxp3^+^ Helios^+^ CTLA4^+^ Nrp1^−^, with effector molecules similar to those of nTregs. Foxp3^−^ iTregs can be further subdivided into various subtypes such as Tr1, which highly expresses the anti-inflammatory cytokine IL-10, Th3 discovered in oral tolerance settings, iTr35 induced by IL-35, as well as Tregs induced by subsets of B cells (Treg-of-B1a, Treg-of-B2), and cytotoxic Tregs (exTregs) [53,54].

### 2.5. Specific CD8^+^ Treg

In 1992, Benvenuto Pernis and colleagues at Columbia University confirmed the immunosuppressive function of CD8^+^ T cells in a murine experimental allergic encephalomyelitis (EAE) model [55]. It was subsequently established that CD8^+^ Tregs are restricted by the MHC class Ib molecule Qa-1 and exhibit surface markers such as KIR (murine Ly49) [55,56]. The differences between CD4^+^ Treg and CD8^+^ Treg are shown in Table 2.

**Table 2 ijms-25-07346-t002:** The key differences between CD4^+^Tregs and CD8^+^ Tregs.

Feature	CD4^+^ Treg	CD8^+^ Treg	References
Thymic Selection	Thymic epithelial cells	Thymic hematopoietic cells	[55,56]
MHC Restriction	MHC class II	MHC class Ib (Qa-1)	
Major Transcription Factor	Foxp3/Helios	Helios	
Antigen	Tissue-specific self-antigens	Stress-related antigens, Hsp60, FL-9 peptide	
TCR Diversity	Polyclonal	Limited	
Surface Markers	CD25, GITR, Nrp-1	CD44^+^CD122^+^Ly49^+^	
Cytokine Dependency	IL-2	IL-15	
Main Function	Suppress excessive immune responses, Maintain tissue homeostasis	Suppress excessive immune responses	

## 3. The Effect of Various Chemokines on Treg Homing in Different Tumors

Chemokines play a crucial role in the immune response to cancer by facilitating the recruitment of Treg cells to the TME (Figure 1). Notably, chemokine receptors, including C-C chemokine receptor type 2, 4, 5, 6, 8, 10 (CCR2, CCR4, CCR5, CCR6, CCR8, CCR10), and C-X-C chemokine receptor type 3, 4 (CXCR3, CXCR4) have been identified as significant factors in the recruitment of Treg cells to the tumor site [57,58,59]. These chemokine pathways facilitate the movement of Treg cells to the TME in response to CC and CXC chemokines, thus influencing the immune environment within the tumor [58,60]. The specific array of chemokine receptors expressed by Treg cells implies their prompt recruitment to inflammatory sites, enabling them to effectively regulate immune responses [61]. Furthermore, the expression of chemokine receptors by Treg cells is dynamic, with changes in trafficking receptors observed in secondary lymphoid tissues, underscoring the importance of these receptors in Treg cell migration and function [62].

### 3.1. The Roles of CC-Chemokine Receptors for TME Recruitment

CCR2 recruits Treg cells to the TME in melanoma and breast cancer [63,64,65,66]. C-C Motif Chemokine Ligand (CCL) 2, a ligand for CCR2, is expressed in breast cancer tissues, promoting the activation and migration of tumor antigen-specific Tregs to tumor sites [64]. CCR4 plays a significant role in Treg recruitment in ovarian cancer, lung cancer, and melanoma [67,68,69]. Studies highlight CCR4’s role in recruiting and activating Tregs near primary tumors, resulting in adverse clinical outcomes [70]. The interaction between myeloid cells and tumor cells triggers the production of CCL22, a CCR4 ligand, facilitating Treg recruitment and the suppression of conventional T-cell responses [70]. CCR5 is crucial for attracting Treg cells to the TME across various cancer types, such as colorectal cancer [71]. Absence of CCR5 reduces Treg infiltration in tumors, delaying tumor progression [71]. In pancreatic ductal adenocarcinoma, inhibiting the CCL5/CCR5 pathway reduces Treg recruitment to the TME [72]. CCR6 impacts hepatocellular carcinoma (HCC) by recruiting Treg cells to the tumor site. Heightened CCR6 expression in tumor-associated Tregs is linked to tumor advancement. CCL20, a CCR6 ligand, attracts CCR6-expressing Tregs, promoting tumor progression and unfavorable outcomes in HCC. The CCL20-CCR6 axis guides circulating Tregs into the TME, fostering tumor progression and worsening prognosis in HCC patients [73,74,75]. CCR8 facilitates Treg migration to the TME, with high CCR8 expression on Tregs infiltrating solid tumors such as breast, lung, and colorectal cancers [76,77,78]. The SRY-box transcription factor 2 (Sox2) promotes the recruitment of Treg cells to TME through elevated expression of CCL1 and CCR8 [79]. The CCL1-CCR8 axis significantly contributes to Treg recruitment and immunosuppressive activities in the TME [80]. CCR8 is distinctly upregulated in tumor-resident Tregs in breast, colon, and lung cancers compared with normal tissues [77,78,81]. CCR10 is vital for recruiting Treg cells to the TME in ovarian cancer [54]. The presence of hypoxia in ovarian cancer triggers the overexpression of CCL28, a ligand specific to CCR10, by tumor cells. This heightened expression of CCL28 in hypoxic conditions fosters the attraction of CCR10-expressing Treg cells to the tumor site, thereby facilitating tumor tolerance and angiogenesis [54,82]. CCL28 released by hypoxic tumor cells recruits CCR10^+^ Tregs, promoting tumor progression and angiogenesis [82]. The CCL28-CCR10 interplay is essential for Treg migration to the tumor vicinity.

### 3.2. The Roles of CXC-Chemokine Receptors for TME Recruitment

CXCR3 and CXCR4 play crucial roles in mobilizing Treg cells to tumor locations in various cancers, including ovarian cancer, hepatocellular carcinoma (HCC), breast cancer, and lymphoma. In ovarian cancer, CXCR3 facilitates the migration of tumor-infiltrating lymphocytes, including Tregs, promoting anti-tumor immune responses [83]. CXCR3 deficiency can prolong Th1-type hypersensitivity reactions [84]. In HCC, Treg recruitment via CXCR3 contributes to the immunosuppressive tumor microenvironment (TME) [85]. CXCR4 is involved in Treg mobilization to the TME in ovarian cancer, breast cancer, and lymphoma [86,87,88]. In basal-like breast cancer, hypoxia increases CXCR4 expression in Tregs, enhancing their recruitment to the tumor site and suppressing the immune response [88]. In ovarian cancer, the CXCR4-CXCL12 axis attracts monocytic myeloid-derived suppressor cells (MDSCs) to the tumor, with PGE2 inducing functional CXCR4 and CXCL12 expression [89]. In lymphoma, CXCR4 recruits Tregs to the tumor mass, possibly through CCL17 and CCL22 [90].

In conclusion, the intricate interaction between chemokines and their receptors is crucial for Treg cell migration to the tumor site. The modulation of TME can impact the efficacy of cancer immunotherapy and disease progression. (Figure 1).

## 4. Survival Strategies and Metabolic Characteristics of Tregs and Their Impact on Function

In the TME, beyond the impact of chemokine receptor expression on the abundant accumulation of Tregs, various cell surface biomarkers and the distinctive metabolic mechanisms of Tregs also play significant roles. Interactions among cells within the TME foster a competitive milieu, contributing to the exhaustion of tumor-infiltrating lymphocytes (TILs). Furthermore, the tumor cells’ heightened glucose utilization yields substantial lactate production, consequently lowering the environmental pH and impairing the cytolytic activity and cytokine production of TILs. Additionally, the TME encompasses factors such as hypoxia, aberrant vascularization, dense stroma, and cancer-associated fibroblasts (CAFs), all of which influence and modify the metabolic pathways and metabolites of TILs.

### 4.1. HIF-1 Impact on Treg Function in Hypoxia

In a hypoxic environment, the hypoxia-inducible factor HIF-1 pathway is activated in solid tumors. In cancer cells or infiltrating immune cells, HIF-1a can recruit Treg by promoting CCL20, transforming growth factor-β (TGF-β), and vascular endothelial growth factor-A (VEGF-A), causing them to migrate to the TME [75,91,92]. There remains considerable debate regarding the relationship between the regulation of HIF-1α and Foxp3 and the suppressive function of Tregs. Research suggests that hypoxia can induce Foxp3 expression by upregulating HIF-1α levels in CD4^+^ T cells, consequently facilitating the differentiation of CD4^+^ T lymphocytes into Tregs [93]. HIF-1α enhances its transcriptional activity by binding to the hypoxia response element located upstream of Foxp3 [94]. This mechanism is implicated in the functional enrichment of Tregs while also stabilizing their activity and suppressive functions. Knockdown of HIF-1α, however, results in a skewing towards Th1 immune response [95]. However, conversely, some studies have indicated that under hypoxic conditions, the deletion of HIF-1α in Tregs enhances their suppressive function. This is attributed to the role of HIF-1α in impeding the entry of pyruvate into mitochondria [96], consequently promoting glycolysis. This results in a reduction in Treg suppressive function and an enhancement in Treg migratory capability. Additionally, the N-terminal 80 amino acids of HIF-1α can bind to the C-terminal of Foxp3, facilitating the degradation of Foxp3 via the prolyl hydroxylases and von Hippel-Lindau protein (PHD-VHL)-mediated ubiquitination degradation pathway [97]. Furthermore, Tregs deficient in HIF-1α are less affected when fatty acid metabolism is suppressed, whereas lipid metabolism is indispensable for Tregs, implying that HIF-1α also plays a crucial role in fatty acid metabolism [96]. Under hypoxic conditions, the activation of HIF-1α has been shown to have diverse effects on Treg generation. While it can promote the recruitment of certain Tregs in specific environments, it may also impair the function of other Tregs.

### 4.2. Reprogramming of Energy Metabolism in Tregs in the TME

#### 4.2.1. Tregs Utilize Lactate to Proliferate

The cellular energy metabolism pathways are crucial for cell survival and function, with cellular metabolic reprogramming being widely recognized as a hallmark of cancer. Tregs undergo reprogramming in the TME to adapt to the low oxygen and high lactate conditions, meeting the increased energy demands and biosynthesis following activation. It is generally agreed that lactate, as a byproduct of tumor metabolism, serves as a nutrient rather than a toxin for Tregs. In the metabolism of Tregs in human liver cancer, lactate uptake is increased through upregulation of monocarboxylate transporter 1 (MCT1). Subsequently, lactate is converted to pyruvate by lactate dehydrogenase and enters the tricarboxylic acid cycle to generate ATP [98]. Moreover, lactate uptake leads to increased expression of PD-1 in Tregs, enhancing their suppressive function while attenuating the efficacy of immune checkpoint inhibitors [98]. Tregs utilize lactate-derived carbon to produce phosphoenolpyruvate, serving as an intermediate in glycolysis to promote Treg proliferation [99].

#### 4.2.2. Fatty Acid Oxidation Can Support Treg Proliferation

In general, pro-inflammatory cells such as Teff and M1 macrophages primarily rely on glycolysis as a rapid energy-producing method. Moreover, the proliferation and inflammatory functions of CD4^+^ Teff cells also largely depend on Glut1 and glycolysis. Inhibiting glycolysis or lacking Glut1 can impair the function of Teff cells in vivo [100]. Unlike other cells, Tregs have a lower glycolytic rate and primarily rely on fatty acid oxidation (FAO) and oxidative phosphorylation (OXPHOS) for cellular energy, avoiding competition with tumor cells for glucose [101].

Key enzymes involved in FAO include CD36, carnitine palmitoyltransferase-1 (CPT-1), adenosine monophosphate (AMP)-activated protein kinase (AMPK), and liver kinase B1 (LKB-1). CD36 mediates the transmembrane transport of fatty acids, facilitating their entry into fatty acid oxidation pathways. In a mouse melanoma model, inhibiting CD36-mediated lipid transport effectively eliminated Tregs, leading to suppressed tumor growth without causing systemic immune instability [8,58,102]. Furthermore, CD36-induced uptake of free fatty acids (FFA) activates the PPAR-β signaling pathway, promoting lactate oxidation to pyruvate for entry into the TCA cycle and enhancing mitochondrial respiration while simultaneously reducing NAD^+^ to NADH^+^H^+^. This process contributes to the immune suppression induced by Tregs [99].

AMPK, a metabolic sensor, can activate CPT-1 [103]. CPT-1 is a key enzyme in FAO, responsible for catalyzing the coupling of long-chain fatty acids with carnitine to produce acylcarnitine. This process enables the transport of fatty acids into mitochondria for oxidation, ultimately generating acetyl-CoA for entry into the tricarboxylic acid (TCA) cycle and inhibiting fatty acid synthesis (FAS). Etomoxir, by pharmacologically inhibiting CPT-1, affects FAO, leading to reduced Treg proliferation [104].

LKB1, as an upstream regulator of AMPK, independently modulates FAO and OXPHOS to regulate Treg proliferation and prevent their differentiation into Th1 and Th17 cells. Additionally, LKB1 is involved in the conversion of mevalonate to cholesterol. Inhibition of HMGCR leads to elevated secretion of IFN-γ and IL-17 by Tregs, while mevalonate and its metabolite, geranylgeranyl pyrophosphate, can inhibit Treg differentiation [105].

#### 4.2.3. The Balance of Treg Proliferation and Functional Stability

Oxidative phosphorylation supports the function and stability of Tregs, while increased glycolysis promotes Treg proliferation but undermines their functional stability. Activated Tregs exhibit elevated mTOR activity and Glut1 levels. The glycolytic metabolism of Tregs is tightly regulated by various metabolic signals, including the glucose transporter GLUT1 and the PI3K-Akt-mTOR signaling network. The PI3K-AKT-mTOR signaling pathway can be activated in Tregs, with Toll-like receptor (TLR) stimulation promoting PI3K-AKT-mTORC1 signaling, as well as the expression of glycolysis and Glut1 [106]. However, defects in lipid phosphatase PTEN or tumor suppressor TSC2 can lead to selective loss or constitutive activation of mTORC1, resulting in an increase in the number of Tregs but impairing their suppressive capacity [32,107]. Stimulation of Foxp3 by CTLA-4 and PD-1 inhibits PI3K-AKT-mTORC1 signaling and glycolysis while promoting oxidative phosphorylation. However, overexpression of Glut1 in Tregs leads to downregulation of Foxp3 and reduces the suppressive capacity of Treg cells. Additionally, genetic deletion of Foxp3 results in the loss of Treg cell suppressive function [108].

Activation of TLR1 and TLR2 on Treg cells further enhances Treg glycolysis and proliferation while reducing Treg suppressive capacity in an mTORC1-dependent manner [109]. Evidence also suggests that targeting mTORC2 via the PI3K-AKT pathway promotes glycolysis by activating glucokinase, thereby inhibiting Treg activity [110].

Key regulators upstream and downstream of the PI3K-Akt-mTOR pathway, such as AMPK, PTEN, and HIF-1a, are also important modulators of Treg glycolysis and the balance between proliferation and suppressive function. During PI3K-AKT activation, PI3K catalyzes the phosphorylation of membrane phospholipid PIP2 to PIP3, serving as an anchor point for AKT [111,112]. The activation of AKT depends on phosphorylation at S473 and T308, with mTORC2 responsible for S473 phosphorylation and PDK1 for T308 phosphorylation [113,114,115]. PTEN counteracts PI3K-AKT activation by catalyzing the reverse reaction, dephosphorylating PIP3 to PIP2 [115]. The deletion of PTEN relieves the inhibitory effect, leading to increased glycolysis in Tregs [32].

Additionally, evidence implicates the involvement of HIF in directing the shift toward glycolysis. Indeed, HIF-1α has been shown to upregulate the expression of pyruvate dehydrogenase kinase 1 (PDK1), which in turn inhibits the mitochondrial enzyme pyruvate dehydrogenase (PDH). As a result, the conversion of pyruvate to acetyl-CoA is hindered, preventing its entry into the TCA cycle. Instead, pyruvate is shunted towards lactate production via glycolysis [116]. At the same time, HIF-1a enhances cellular glucose uptake by increasing the expression of GLUT1 encoded by SLC2A1 and GLUT3 encoded by SCL2A3 [117,118,119]. Furthermore, HIF upregulates the expression of all enzymes in glycolysis, including phosphofructokinases (PFKL and PFKP), hexokinases (HK1/2), aldolases (ALDA/ALDC), glyceraldehyde-3-phosphate dehydrogenase (GAPDH), phosphoglycerate kinase1 (PGK1), enolases (ENO1/2), and pyruvate kinases M (PKM) [120,121,122,123,124,125].

Inhibition of mTOR activity can lead to the activation of autophagy, and Tregs with defective autophagy have been shown to exhibit high levels of mTORC1-Myc pathway activity and glycolytic activity [126,127,128,129]. Additionally, autophagy also serves to suppress mTORC1 activity in Tregs. Deficiency in the autophagy gene Atg7 leads to heightened mTORC1 activity and glycolysis in Tregs. Furthermore, deletion of the PP2A subunit PP2AA, a natural inhibitor of mTORC1, results in increased mTORC1 activation and glycolysis in Tregs, ultimately impairing Treg suppressive function [128,129].

#### 4.2.4. Therapeutic Potential of Targeting Treg Metabolic Pathway

In recent decades, immunotherapy has significantly advanced the treatment of cancer, including the development of immune checkpoint inhibitors (ICIs) such as anti-CTLA-4 and anti-PD-1/PD-L1 monoclonal antibodies. However, tumor-resident immunosuppressive regulatory T cells (Tregs) have proven to be a substantial barrier to the effectiveness of ICI therapy. Targeting immunosuppressive molecules expressed by Tregs through the strategy of blocking chemokine/chemokine receptors presents a promising approach for cancer treatment. Nonetheless, depleting Tregs can often lead to severe autoimmune reactions. In this context, exploring the metabolic pathways of Tregs within the TME and implementing adaptive metabolic interventions may offer novel therapeutic avenues.

A monoclonal antibody targeting CD36 was demonstrated in a melanoma xenograft mouse model, which specifically blocks the uptake and metabolism of fatty acids. This approach effectively suppresses the accumulation of Tregs in tumors without reducing the levels of systemic Tregs and preventing severe autoimmune reactions after treatment [102]. Furthermore, the TLR8 agonists have been shown to reverse the inhibitory function of Tregs by inhibiting mTOR-HIF1α-induced glucose metabolism in tumors, which was confirmed in ovarian cancer and melanoma models [130,131]. The evidence above suggests that targeting the metabolic pathway of Tregs may have promising potential for cancer treatment.

## 5. Immunosuppressive Mechanism of Tregs

As a subset of lymphocytes capable of suppressing immune responses, Tregs seem to play a crucial role in maintaining immune homeostasis and mediating peripheral tolerance.

Current evidence suggests that Tregs have the capability to suppress the function of various immune cell populations, including CD4^+^ and CD8^+^ T cells, B cells, NK cells, and antigen-presenting cells (APCs), such as DC, monocytes, and macrophages [8,132,133]. Tregs exert suppression on effector T cells’ proliferation and cytokine secretion, concurrently promoting B cells’ unresponsiveness and impeding antibody production [134,135]. Furthermore, Tregs inhibit the expression of co-stimulatory molecules, antigen-presenting molecules, and inflammatory cytokines in APCs, thus attenuating their ability to stimulate T-cell responses [132,133,136,137]. Tregs upregulate the indolemine 2,3-dioxygenase (IDO) pathway, which leads to the degradation of tryptophan. The IDO pathway is activated through the reverse signaling of CTLA4 on dendritic cells, resulting in the expression of IDO and subsequent tryptophan metabolism into the kynurenine pathway, which inhibits T cell activity and induces T cell apoptosis [138,139]. Moreover, Tregs disrupt Il-2 receptor signaling, hindering DC activation and subsequently suppressing Teff proliferation. IL-2 also influences the expression of perforin and granzyme genes in CD8^+^ T cells [140]. Thus, Tregs exert influence across all phases of the immune response.

In summary, the mechanisms underlying Treg-mediated suppression of target cells can be delineated as follows (Figure 2): (1) Selective expression of cell surface inhibitory receptors, such as CTLA-4, TIGIT, LAG-3, PD-1, and TIM-3; (2) Direct cell-to-cell contact, utilizing mechanisms such as granzyme B and perforin or Fas/Fasl for target cell dissolution; (3) Secretion of immunosuppressive cytokines, including TGF-β, IL-10, IL-35, etc.; (4) Induction of IL-2 depletion within the microenvironment, resulting in “IL-2 sink”; (5) Modulation of the TME through metabolic alterations, such as facilitating adenosine accumulation and competing for nutrients, thereby orchestrating T cell polarization and various biological processes; (6) Utilizing extracellular vesicles (EVs) as a novel suppressive mechanism, allowing for cell-contact independent and targeted immune modulation [141,142].

## 6. The Potential Anti-Tumor Role of Tregs

As our understanding of Tregs deepens, the initial view that Tregs always lead to poor tumor prognosis has been found to be biased. Particularly in the later stages of cancer, Tregs have been discovered to possess various potential anti-tumor mechanisms. In studies of colorectal cancer, HNSCC, and gastric cancer, Treg infiltration has been positively correlated with extended survival times in cancer patients, suggesting that Tregs can play a proactive role in suppressing tumor development [11,13,14,143,144]. This effect is related to the stability of Tregs, their subgroup transformation, and their ability to downregulate inflammation that promotes tumor progression.

### 6.1. Th-like Treg Induce Inflammation in TME

Under inflammatory conditions, CD4^+^CD25^high^CD127^(low/−)^ Treg cells exhibit plasticity [145]. This means that in addition to expressing Foxp3, they acquire lineage-defining transcription factors and are referred to as Th-like Treg cells [146,147]. This enables them to potentially adopt functions associated with other T helper cell subsets while maintaining some regulatory characteristics.

#### 6.1.1. Th1-like Tregs

Under the influence of IL-12, Foxp3^+^ Tregs can differentiate into a Th1-like phenotype, marked by the expression of the Th1 lineage-specific transcription factor T-bet (Tbx21). This phenotypic transformation of Tregs into Th1-like Tregs is facilitated by both IFN-γ and IL-12, which stimulate the expression of T-bet within these cells [147,148]. Consequently, these cells assume a T-bet^+^IFN-γ^+^Foxp3^+^ profile, exhibiting decreased suppressive capabilities while increasing IFN-γ secretion. This adaptation enhances their participation in pro-inflammatory immune responses [149,150].

The primary pathway through which Th1-like Tregs secrete IFN-γ involves the activation of the PI3K/AKT/Foxo1/3 signaling pathway. Blocking this pathway can inhibit IFN-γ production and restore the immunosuppressive capabilities of Tregs [151]. In human papillomavirus (HPV)-driven oropharyngeal squamous cell carcinoma (OPSCC), a substantial presence of conventional Tregs that co-express T-bet and high levels of Foxp3 (Foxp3^high^ Tregs) has been identified. Interestingly, a high frequency of T-bet^+^ Tregs correlates with an extended disease-specific survival period. This correlation may reflect their number as indicative of substantial infiltration by CD8^+^ and type 1 effector T cells [152], suggesting that T-bet^+^ Tregs may play a complex role in modulating the immune response within the TME.

In the dextran sulfate model of colitis, Th1-like Tregs are upregulated, and the expression of T-bet and IFN-γ in Tregs precedes the accumulation of conventional Th1 cells. Furthermore, the expression of T-bet within Tregs is a necessary condition for the development of colitis. Tregs lacking T-bet expression exhibit reduced IFN-γ production, demonstrating that at certain stages of inflammation, Tregs may contribute to the inflammation rather than suppress it [152].

#### 6.1.2. Th17-like Tregs

Another controversial subset of Th-like Tregs is the Th17-like Treg, characterized by the phenotype IL-17^+^RORγt^+^Foxp3^+^. The generation of this Th17-like Treg subset is driven by a combination of cytokines, including IL-6 and TGF-β, which promote the expression of RORγt and IL-17 within Tregs [153,154,155].

The role of IL-17 in tumor immunology is quite complex, with evidence suggesting both tumor-promoting and tumor-suppressing effects, leading to contradictory views. On one hand, some researchers propose that IL-17, particularly induced by microbes, can promote tumor progression. For instance, IL-17 is known to inhibit the infiltration of CD8^+^ T cells and increase the infiltration of myeloid-derived suppressor cells (MDSCs) within tumors [156,157]. On the other hand, there is evidence suggesting that IL-17 can enhance immune cell infiltration into tumors via various mechanisms. IL-17 is able to upregulate inflammatory chemokines such as CXCL9, CCL2, CXCL10, and CCL20, which coordinate the recruitment of T cells, NK cells, and DCs [158,159]. Moreover, IL-17A boosts cytotoxic responses by enhancing the expression of cytotoxic molecules such as TNF-α, IFN-γ, perforin, and granzymes [160,161]. It also activates receptors like NKp46, NKp44, NTB-A, and NKG2D, which enhance the cytotoxicity of NK cells [158]. IL-17 has also been shown to inhibit the growth of hematopoietic tumors such as mastocytoma and plasmacytoma by enhancing the activity of cytotoxic T lymphocytes (CTLs) [162]. IL-17 stimulates macrophages to produce IL-12 associated with tumor-specific cytotoxic T lymphocytes (CTLs), and it upregulates the expression of co-stimulatory molecules, MHC class II antigens, and allogeneic stimulatory capabilities in dendritic cell precursors, thereby promoting their maturation [163,164]. In mice models, a deficiency in IL-17 is associated with reduced IFN-γ levels in tumor-infiltrating NK cells and T cells [165]. In ovarian cancer, high expression of IL-17 is positively correlated with greater infiltration of cytotoxic IFN-γ^+^ CD4^+^ and IFN-γ^+^ CD8^+^ T cells [166]. In this context, IL-17 secreted by Th17-like Tregs may contribute positively to the prognosis of tumor patients [166,167].

### 6.2. IL-10 Secreted by Pro-Inflammatory Tregs Has Potential Anti-Tumor Effects

IL-10, as an anti-inflammatory cytokine, suppresses inflammatory cytokines such as IL-6 and the IL-12/23 complex to inhibit macrophage activation and the pro-inflammatory Th17 cell responses, confirming the anti-inflammatory role of IL-10 [168]. The anti-inflammatory effects of IL-10 have led to the hypothesis that IL-10 might undermine immune responses against cancer. However, mice lacking IL-10 signaling spontaneously develop tumors at a high rate [169]. Moreover, IL-10 secreted by Tregs plays a crucial role in promoting CD8^+^ T cell activation, leading to the production of granzyme B (GzmB) and IFN-γ, which are key components in the cytotoxic response against tumors (Figure 3) [170,171]. Research indicates that IL-10 produced by Tregs interacts with IL-10 receptors (IL-10R) on CD8^+^ T cells, enhancing the activation and infiltration of Tregs. This interaction also triggers the phosphorylation of STAT1 and STAT3, leading to the production of granzymes and IFN-γ. Additionally, IL-10 can stimulate the production of IFN-γ, which in turn increases the expression of major histocompatibility complex (MHC) molecules, enhancing the presentation of tumor antigens [169].

Furthermore, research shows that IL-10 possesses immunostimulatory properties for CD4^+^ and CD8^+^ T cells as well as NK cells, leading to an increase in the production of IFN-γ [172]. In the context of using IL-10 to treat Crohn’s disease, it has been observed that IL-10 induces the pro-inflammatory cytokine IFN-γ [173].

### 6.3. Tregs Suppress Inflammation That Promotes Tumor Growth and Metastasis

Various chronic inflammations in vivo can induce tumorigenesis, such as hepatitis B virus (HBV) and hepatitis C virus (HCV) mediated liver inflammation leading to hepatocellular carcinoma, human papillomavirus (HPV) infection associated with cervical cancer, and inflammatory bowel disease associated with colorectal cancer [174]. Inflammatory cells and their produced mediators can promote tumor angiogenesis, facilitating tumor cell proliferation and invasion [175]. Mast cells (MCs) contribute to tumor immune evasion by shaping the inflammatory TME [176]. Conversely, Tregs were able to inhibit this process, thereby partially improving prognosis. MCs originate from hematopoietic precursors in the bone marrow, and the release of matrix metalloproteinases (MMPs), including tryptase and chymase during their degranulation process plays a crucial role in the degradation of extracellular matrix (ECM) within the TME [177]. Furthermore, several studies indicate that mast cells express pro-angiogenic compounds, participating in driving tumor angiogenesis and growth, such as vascular endothelial growth factor (VEGF)-A, VEGF-B, fibroblast growth factor, histamine, heparin, and stem cell factor [178,179,180,181]. Tregs employ diverse mechanisms to hinder mast cell differentiation and impede their degranulation. Notably, the OX40/OX40L pathway emerges as a critical regulator in modulating IgE-mediated mast cell degranulation and mitigating the release of mast cell effector molecules [182]. Through interactions with mast cells mediated by the OX40/OX40L axis, Tregs effectively suppress the influx of calcium ions subsequent to FcεRI activation, thereby attenuating FcεRI-dependent mast cell degranulation (Figure 3) [183]. Simultaneously, IL-10 secreted by Tregs can bind to receptors on mast cells, effectively inhibiting the expression and signaling transduction of the IgE receptor FcεRI within mast cells, consequently suppressing mast cell degranulation [184,185,186]. Introducing stable Tregs into mice with polyps can reduce the size of polyps and decrease mast cell infiltration. This process requires the involvement of IL-10, further supporting this regulatory mechanism [187].

### 6.4. Inhibition of Treg on Tumor-Associated Macrophages (TAM)

Tumor-associated macrophages (TAMs), as crucial constituents of the TME, are recruited from peripheral blood monocytes via chemotactic factor signaling [188]. The quantity of TAMs infiltrating the tumor correlates positively with cancer cell invasion depth, microvessel density, and cyclooxygenase-2 (COX-2) expression [189]. The augmented presence of TAMs correlates with the survival outcomes across several human malignancies [190], with heightened infiltration of M1-type TAMs linked to improved overall survival among patients with gastric cancer [191], while elevated infiltration of M2-type TAMs serves as an adverse prognostic indicator in this context [192,193]. M2-type TAMs secrete pro-angiogenic factors such as VEGF, PDGF, and MMP, thereby enhancing tumor angiogenesis and vascular formation. Furthermore, they promote tumor cell invasion and metastasis by secreting proteases that degrade the extracellular matrix and by inducing tumor cell epithelial-mesenchymal transition (EMT). Additionally, M2-type TAMs, characterized by the expression of markers such as CD163 and CD206, enable them to mediate immune suppression within the TME, further supporting tumor progression [194,195].

Tregs can suppress LPS-induced monocyte CD14 retention by engaging in the pro-apoptotic mechanism involving the Fas/FasL pathway, leading to monocyte apoptosis, thus inhibiting TAM origination [196]. In the intestinal mucosa, Treg cells can inhibit CX3CR1^+^ macrophages by binding to MHC class II via the immune checkpoint receptor latent activation gene-3 (LAG-3) [197]. Additionally, type 1 regulatory T cells (Tr1), characterized as CD4^+^Foxp3^low^CD49b^+^LAG-3 T cells, demonstrate elevated secretion of IL-10 and TGF-β. In the context of malignant melanoma, Tr1 cells exhibit cytotoxicity to TAM by releasing granzyme B and perforin [198]. Interestingly, the cytokines such as IL-10 and TGF-β were also secreted by MSDCs, which can promote the development and function of Tregs, suggesting the complex significance of crosstalk between MSDCs and Tregs [199,200].

### 6.5. The Inhibition of Tregs on Tumorigenic Inflammation Caused by Th2/Th17 Cells

Several studies have elucidated the contributory role of Th2/Th17 cells in tumor growth processes. In pancreatic cancer, high Th2 cell infiltration is associated with reduced survival rates [201]. In hypopharyngeal cancer, Th2/Th17 cytokine expression increases with advancing clinical stage [202]. Patients with renal cell carcinoma have higher proportions of Th2 and Th17 cells in peripheral blood compared with healthy volunteers, with this elevation intensifying as tumor staging progresses [203]. These findings suggest that Th2 and Th17 cells are involved in tumor development. Th2 cytokines such as IL-4, IL-10, and IL-13 can alternately promote M2 polarization of TAMs [204,205,206,207]. Furthermore, IL-4 can induce protease activity in M2-type TAMs, thereby promoting tumor invasion and metastasis [208,209,210].

Tregs play a critical role in limiting the oncogenic inflammation induced by Th2/Th17 cells (Figure 3). Tregs achieve this by expressing CTLA-4 and depleting IL-2, suppressing Th2 differentiation and promoting their apoptosis [211]. Tregs also release immunosuppressive molecules such as IL-10 and TGF-beta, which inhibit the pro-tumorigenic inflammatory response induced by Th2/Th17 activation [212]. Research suggests that Treg’s ability to suppress cancer development triggered by microbial stimuli heavily relies on IL-10, which can downregulate pro-inflammatory cytokines like IL-6, thereby directly inhibiting Th2 cell activation [213]. IL-10 can also induce STAT3 activation in Tregs, which plays a pivotal role in suppressing Th17 responses [214,215]. Elevated STAT3 expression in Tregs correlates with increased levels of inhibitory surface molecules, cytokine receptors, and chemokine receptors, inhibiting Th17 activity and downregulating soluble mediators involved in Th17 differentiation [214]. Additionally, the interaction between IRF4 and Foxp3 in Tregs regulates the expression of IL-10 and Gzmb, initiating a Th2-specific inhibitory mechanism [216]. Zhang et al. found that Treg depletion in pancreatic cancer leads to increased pathological CD4^+^ T cells, differentiation of inflammatory fibroblast subpopulations, and myeloid cell infiltration through CCR1, accelerating tumor progression [217].

### 6.6. Treg Destabilization and Reprogramming Implicate the Cancer Immunotherapy

The stability of Tregs is defined by the persistent expression of Foxp3, facilitated by the hypomethylation of its conserved non-coding sequence 2 (CNS2) [218]. CNS2 acts as a crucial transcriptional enhancer for Foxp3, binding transcription factors such as Foxp3 itself, STAT5, and the cAMP response element-binding protein (CREB) [219,220,221]. In stable Tregs, CNS2 is concurrently bound by AML1/Runx1 and Foxp3, triggering DNA demethylation in this region, known as the Tregs-specific demethylation region, which is essential for maintaining Foxp3 expression [222]. In iTregs, high methylation of the CNS2 region prevents the recruitment of crucial transcription factors, leading to unstable Foxp3 expression [223,224]. Additionally, Foxp3 expression is regulated by IL-2; when IL-2 binds to its receptor IL-2R, it activates downstream STAT5, which helps maintain Foxp3 expression. When mature Tregs divide in an IL-2-limited environment, Foxp3 expression decreases, leading to unstable Tregs or ex-Tregs [221,225,226]. Compared with Tregs, ex-Tregs not only exhibit lower Foxp3 expression but also upregulate CD16, CD56, CD127, and cytotoxic and inflammatory genes such as TBX21, NKG7, CCL3, CCL4, and CCL5 [227,228]. The cytotoxicity of ex-Tregs has been functionally validated in cell-killing assays and CD107a degranulation experiments [228,229,230]. Additionally, in colorectal cancer, variations in cytokines secreted by intestinal microbiota, specifically IL-12 and TGF-β, lead to different ratios of Foxp3^high^ Tregs and Foxp3^low^ Tregs. Foxp3^low^ Tregs lose their immunosuppressive abilities and instead secrete IL-12 and IFN-γ, which correlates with a better prognosis for patients [231]. In the models of melanoma, blocking PTEN by VO-OHpic following chemotherapy or immunotherapy has been shown to induce the generation of ex-Tregs, which effectively disrupts the ability of tumors to re-establish a suppressive TME [232]. This highlights the importance of accurately assessing Tregs and ex-Tregs (Figure 4A).

Another distinct type of Tregs, known as “fragile Tregs”, differs from ex-Tregs. Unlike ex-Tregs, “fragile Tregs” maintain Foxp3 expression but lack surface expression of the receptor Nrp1 (Figure 4B). The majority of tumor-infiltrating Tregs express Nrp1, which interacts with its ligand, semaphorin 4A (Sema4A). This interaction inhibits Akt phosphorylation within the cell via PTEN, promoting the nuclear localization of the transcription factor Foxo3 and enhancing the suppressive function of Tregs [233]. Additionally, the binding of Nrp1 to the vascular endothelial growth factor (VEGF) facilitates the migration of Tregs to inflammatory sites or TME, which is a prerequisite for Tregs to exercise their functions [234]. Nrp1 works synergistically with Foxp3 and CTLA-4 to enhance the suppressive activity of Tregs. Specifically, Nrp1-mediated interactions between Tregs and dendritic cells (DCs) can be prolonged, limiting the proximity of effector T cells to antigen-presenting cells (APCs), thereby reducing their activation and proliferation [235].

**Figure 4 ijms-25-07346-f004:**
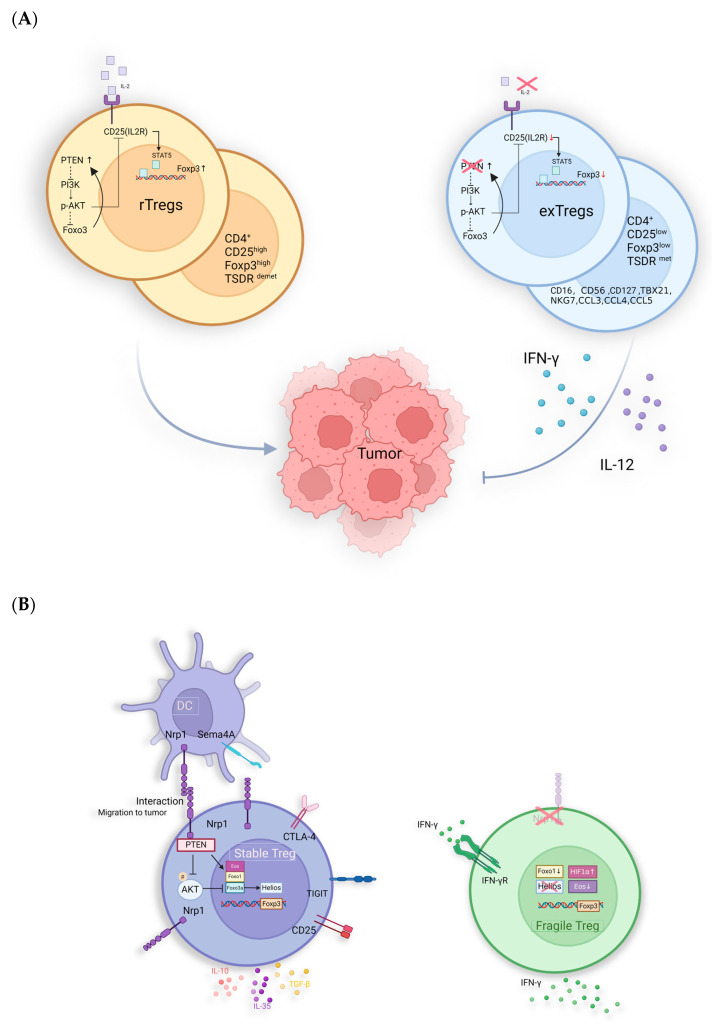
Potential anti-tumor immunological mechanisms of untypical Tregs: (**A**) PTEN maintains the stability of Tregs via inhibiting the activation of PI3K/Akt to stabilize the expression of CD25 and Foxp3. The IL-12-restricted TME induces reduced expression of Foxp3, CD25, and PTEN, leading to the generation of ex-Tregs with pro-inflammatory functions. (**B**) Fragile Tregs: absent Nrp1 and express Foxp3 while secreting IFN-γ with anti-tumor activity. Helios is also essential for the stability of Tregs, with its loss leading to an impairment of their suppressive function [236]. In studies related to ATL, Helios knock-down can enhance the drug sensitivity of ATL cells [237]. The absence of Helios in Fragile Tregs further suggests its significant role in tumor promotion.

In both mouse models and humans, the absence of Nrp1 leads to fragile Tregs that secrete IFN-γ, inhibiting tumor growth [238]. Enhanced IFN-γ signaling is a hallmark of “hot” tumors, which are characterized by an immunoinflammatory phenotype [239]. In melanoma cells, low-level expression of ERK signaling is essential for tumor cell survival. IFN-γ can lead to the overactivation of the ERK pathway, thereby inducing cell death [240]. Additionally, IFN-γ can act on nearby Tregs, causing them to convert to an unstable phenotype [241]. This effect is further amplified in hypoxic environments due to the upregulation of HIF1α [242]. In patients with metastatic melanoma and HNSCC, the fragile Treg subgroup is associated with a better prognosis. Moreover, Nrp1 is not essential for suppressing autoimmunity and maintaining immune homeostasis [243,244]. Therefore, inducing the conversion of Tregs to an Nrp1^−/−^ phenotype presents a promising therapeutic direction [241]. This approach could counteract the tumor-protective immune responses of Tregs, improving prognosis without triggering excessive autoimmune reactions [243,244]. Additionally, the absence of Foxo1 and Eos, as well as the activation of the IFN-1/IFNAR1 pathway, can also induce the conversion of Tregs to fragile Tregs, which then produce IFN-γ, mediating better prognosis [245,246,247]. However, the proportion of fragile Tregs in tumors and their therapeutic utility remains controversial.

## 7. Conclusions

In recent years, studies have observed an increased frequency of Tregs in the peripheral blood and tumor tissues of patients with various cancers, including lung cancer, breast cancer, ovarian cancer, colorectal cancer, gastric cancer, head and neck cancer, renal cell carcinoma, liver cancer, leukemia, lymphoma, and melanoma. Many reports suggest that elevated Tregs in tumor tissues correlate with reduced survival rates and treatment responses. However, the dual role of Tregs in tumor development explains why, in some cancers, higher Treg frequency is associated with favorable outcomes. Particularly in hematological malignancies, an increased number of Tregs predicts a favorable prognosis. For instance, in patients with follicular lymphoma, an increase in the number of tumor-infiltrating Foxp3^+^ T cells correlates with improved survival rates and decreased follicular lymphoma counts [248]. Similarly, in patients with Hodgkin’s lymphoma, a decrease in Foxp3^+^ T cells is associated with poorer survival rates [249].

Tregs play a dual role in tumor occurrence and development. On one hand, they inhibit systemic inflammatory reactions resulting from excessive activation of anti-tumor immunity, which is essential for maintaining immune homeostasis. On the other hand, in the late stages of tumors, Tregs can downregulate anti-tumor immune responses, thereby suppressing tumor cell death. In the early stages, Tregs can also suppress inflammation, which is favorable for tumor development. For example, RORγT^+^ Helios^−^ Tregs, induced by gut microbes, respond to tissue damage-induced IL-33 and can restrict tissue damage and prevent tumorigenesis during colitis [250]. Accumulation of Foxp3^+^ Tregs in head and neck carcinomas (HNC) has been shown to have a favorable impact on patient prognosis [251,252]. This is likely due to their varied functions within distinct TMEs, influencing their activities across different tumor regions. In colorectal cancer (CRC), prognostic disparities are observed: Tregs within the tumor mesenchyme are linked to better outcomes, while those within the tumor nests and margins are associated with adverse patient prognoses [253,254]. Additionally, significant differences in Treg function and prognosis are observed across various tumor stages. A retrospective analysis in gastric cancer reveals that high Foxp3^+^ Treg infiltration in stages I–II is associated with higher five-year survival rates, contrasting with poorer outcomes in stages III–IV [255]. Such variations may stem from alterations in Treg functional states across diverse TME. The intricate TME and Treg functional diversity dictate their developmental trajectory and impact on tumor prognosis and therapeutics. Although speculative, the dual function of Tregs in tumor development requires further experimental verification. Refining our understanding of Tregs’ roles in tumor development will help us move beyond a one-sided view of Tregs and guide further research on how to utilize them as a therapeutic strategy for tumors.

## Figures and Tables

**Figure 1 ijms-25-07346-f001:**
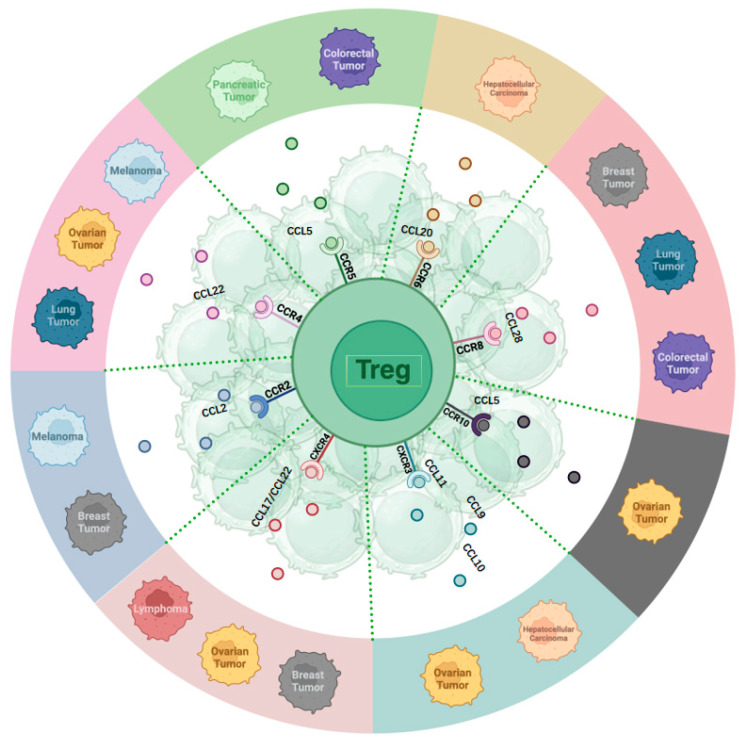
Chemokines receptors and their primary ligands affecting Treg homing in various tumors.

**Figure 2 ijms-25-07346-f002:**
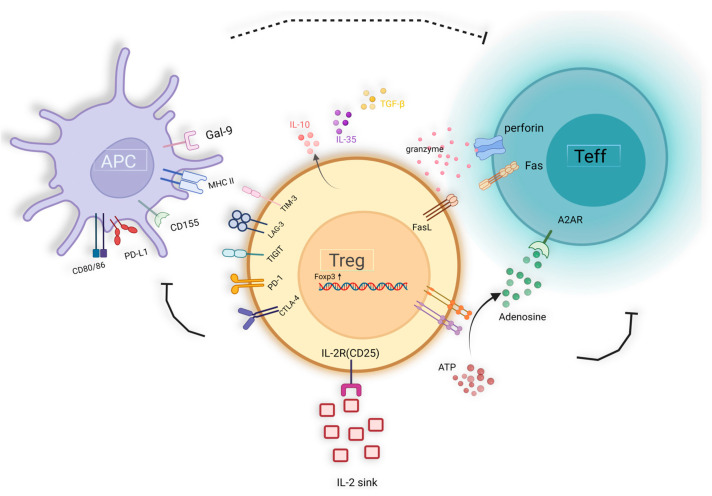
The immunosuppression mechanism of Treg cells.

**Figure 3 ijms-25-07346-f003:**
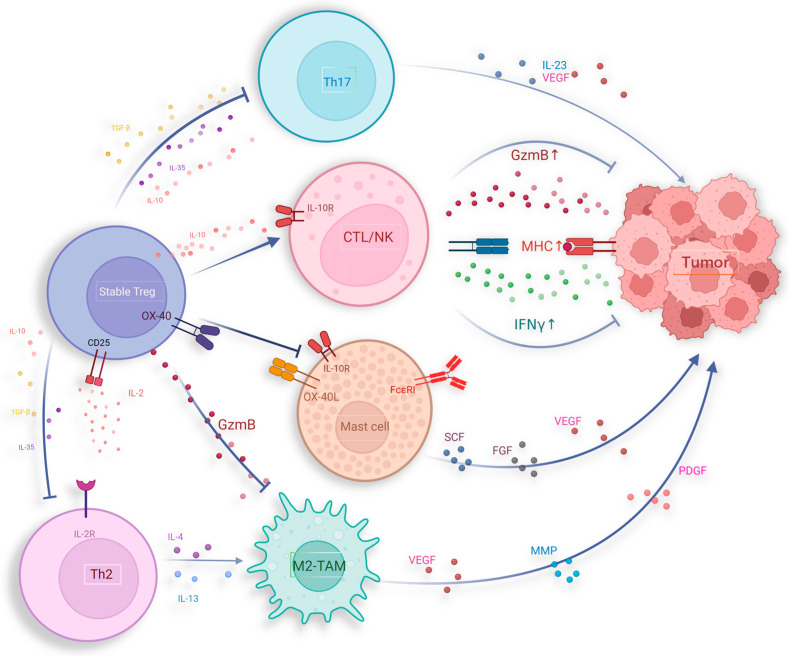
Potential anti-tumor activities of Tregs, including suppressing pro-tumor inflammatory responses induced by Th2, Th17, MC, and M2-TAM.

## Data Availability

Not applicable.

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
