# Peer review of "Complex Role of Regulatory T Cells (Tregs) in the Tumor Microenvironment: Their Molecular Mechanisms and Bidirectional Effects on Cancer Progression"

_ijms, 2024, doi:10.3390/ijms25137346_

Round 1

Reviewer 1 Report

Comments and Suggestions for Authors

This manuscript provides a comprehensive overview of the complex roles of Tregs within the TME, elucidating cellular communications, signaling pathways, and their impacts on tumor progression. In addition, the authors highlight the potential anti-tumor role of Tregs and their functional molecules interactions.

Specific comments:

1)     Figure 1 legend: sine the authors illustrate chemokine receptors, the title should be “chemokines receptors affecting Treg homing in various tumors”.

2)     Paragraph 2.1 “The classification of CD4+ Treg subpopulation based on CD45RA”. CD45RA should be better introduced and explained in the text.

3)     Line 293: acronym should be checked.

4)     Paragraph 6.4: while the interaction between Treg and TAMs is well explained, could be nice to highlight the crosstalk between Tregs and MDSC in TME.

5)     English editing and Grammar revision is required (See line 580-581).

Comments on the Quality of English Language

English editing and Grammar revision is required (for example, see line 580-581).

Author Response

Dear Reviewer 1,

Thank you for your valuable suggestions. Here are our responses to each of your points:

  1. We have changed the title of Figure 1 from "Figure 1. Chemokines affecting Treg homing in various tumors" to "Figure 1. Chemokine receptors and their primary ligands affecting Treg homing in various tumors." This is indeed a more accurate description.
  2. Regarding paragraph 2.1, "The classification of CD4+ Treg subpopulation based on CD45RA", additional information was added in line 91-95.
  3. The term "PHD-VHLU-mediated" in line 293 has been changed to "prolyl hydroxylases and von Hippel-Lindau protein (PHD-VHL)-mediated" as mentioned in line 279. Thank you for pointing this out.
  4. We have added an explanation about the "crosstalk between Tregs and MDSC in TME" between lines 572 and 575.
  5. We changed the wording “Additionally,M2-type TAMs, characterized by the expression of markers such as CD163 and CD206,which enable them to mediate immune suppression within the TME, further supporting tumor progression” in line 561-564, and misspelling of "inflammatory." This has been corrected.

Thank you once again for your thorough review and helpful suggestions.

Reviewer 2 Report

Comments and Suggestions for Authors

This manuscript is truly informative and also relatively easy to read - despite the complex topic and the broad spectrum of molecular and cellular interactions that are outlined. 

Most importantly, the review is also understandable for the non-expert who is searching for an accessible introduction into the topic. These factors should contribute to the broader acceptance of this review as an introduction and (hopefully) will receive a considerable number of citations.  There is little left for criticism from my side, and mostly covers details that could be possibly still improved. As a whole, I think the manuscript is more or less publishable in current version. 

1) there are already several figures in the manuscript, and all of them are clear and not overly complex. They also are "to the point", meaning, they have a clear message and dont overlap. Nevertheless, the characterization of Tregs and distinction to other, related immune cell populations could also benefit from an additional figure (matching text in lines 88-100, and then again lines 103 - 125), or a schematic presentation of the relationship of these cell types and their precursors when they develop and  mature into functional cells. Something like a pedigree cartoon would be helpful. 

I always find this classification issue extremely complex and its hard to remember the marker profiles characteristic for these types and subtypes of cells. Especially, since subclassification gets more and more detailed.

Cytokine features from the following paragraph are covered in a figure. 

The classification issue is really important as it is the dominant topic throughout the1st pages of the manuscript. ANd its even more relevant since text sections that mentione complex marker profiles  (like this example: "phenotype CD4+ CD25+ Foxp3+ Helios+ CTLA4+ Nrp1+. Their effector molecules include TGF-β, IL-10, CTLA-4, IL-35, LAG3, and LAP") are simply difficult to read (lines 146-147). It is not easy to focus on such texts, with tons of numbers; they are not reader-friendly. 

I know table 1 covers this issue... but, tables are also a bit "dry" to read, especially when they spread over almost 2 pages, like in this case. 

The entire next paragraph is dedicated to the role of chemokines and cytokines in T-cell differentiation and functional activation/inactivation. This is covered by Figure 1. But - it looks incomplete and only shows the receptors, plus the tumour types in which this may play a role. Why are the ligands not mentioned in Figure 1? I would find this much more relevant and interesting than mentioning the tumour types. Its also likely not "the last word" and research will certainly find additional tumor subtypes in which Tregs are involved. This, this figure will not age very well. A figure with the ligands will probably age a lot better. 

The next few paragraphs look beyond the "usual suspects" (= chemokines) and also cover other molecular pathways and stress conditions, including hypoxia and metabolic changes (lactate, fatty acid consumption) in tumor tissues.  This is really nice, since it may not be covered in much detail in existing reviewed articles. 

Important is the link to potential or established targets for immunotherapy, such as IDO and CTLA4. This section could almost be expanded as this is one of the critical issues that attracts readers from different fields (oncology, immunology, cancer-immunotherapy, pharmacology, etc). 

The same applies of course for section 6: This is the paragraph that will draw attention and interest from readers and mot likely result in this article to be cited frequently - provided it hits the right tune. The many Treg subtypes are again introduced in the form of text-only subparagraphs but this time, there is also a nice, comprehensive figure (Fig. 3) that goes with this somewhat dry description. 

Otherwise, I have to say that the manuscript is very detailed, but not too detailed, and it requires the reader to spend significant time on the paper to understand. From my understanding, the references are mostly up-to-date and representative. As this is a rapidly evolving field of research, its important not to start with outdated references. 

Author Response

Dear Reviewer 2,

Thank you for your very encouraging comments and numerous valuable suggestions. Here are our responses to each of your points:

  • As a summary of the text explaining the differentiation between Tregs and other immune cell populations in the first half, we have used Table 1. Regarding the issue with Table 1: to prevent the table from being spread across two pages, we have rearranged the text and table so that they are both consolidated on page 4. We believe this makes it easier to understand compared to the original manuscript. Additionally, regarding your suggestion that "a schematic presentation showing the relationship when precursor cells develop and mature into functional cells would be helpful," we feel that including this would lead to an excessive number of references and may blur the focus of this review. Therefore, we hope you will understand our decision to omit this item.
  • Regarding your comment on Figure 1: "This is covered by Figure 1. But - it looks incomplete and only shows the receptors, plus the tumor types in which this may play a role. Why are the ligands not mentioned in Figure 1?" we added the major ligands in Figure 1, and change the title to “ Chemokine receptors and their primary ligands affecting Treg homing in various tumors.”. This review focuses on the relationship between cancer and Tregs, so the figure is intended to summarize and illustrate how these factors generally influence the differentiation of T cells and Tregs.
  • We are honored by your compliment: "This is really nice, since it may not be covered in much detail in existing reviewed articles." We also aim to provide a "new perspective."
  • "Important is the link to potential or established targets for immunotherapy, such as IDO and CTLA4." >>> We fully agree. In fact, adult T-cell leukemia (ATL), which we are dealing with, is a typical Treg-type tumor, and this significantly complicates its treatment. We recognize the importance of antibody therapeutics, such as anti-CCR4 antibody (mogamulizumab) and anti-CD30 antibody (brentuximab vedotin). However, as this review presents an almost opposite perspective, we have intentionally chosen not to address that aspect in this article. Instead of that, we added the section 4.2.4 to explore the new therapeutic potential targeting metabolism of Treg and a new evident in 6.6 to discuss the novel therapy to combined chemotherapy/ immunotherapy with alter the Treg cell programming.
  • Section 6, as you pointed out, represents the core part of this paper. Until now, the pro-tumor aspects of Tregs have been the focus, but we anticipate that more and more examples will emerge demonstrating that their function of suppressing excessive immune activation can also lead to anti-tumor activity within the TME.
  • "From my understanding, the references are mostly up-to-date and representative. As this is a rapidly evolving field of research, it’s important not to start with outdated references." Thank you for your insightful suggestion. However, we also believe it is important to show respect to the researchers who made the original discoveries. Balancing this with the latest knowledge is indeed crucial. Thank you for your valuable feedback.

We sincerely appreciate your very thorough and constructive comments.
